# Reducing alcohol consumption in UK armed forces veterans: Feasibility of using personalized push notifications with AI

Daniel Leightley[1]*, Charlotte Williamson[1], Iain J. Marshall[1], Vasa Curcin[1], Roberto J. Rona[2], Dominic Murphy[2,3], Nicola T. Fear[2†], Laura Goodwin[4]

**1** Department of Population Health Sciences, School of Population Health Sciences, King's College London, London, United Kingdom, **2** King's Centre for Military Health Research, King's College London, London, United Kingdom, **3** Combat Stress Centre for Applied Military Health Research, Leatherhead, United Kingdom, **4** Faculty of Health and Medicine, Lancaster University, London, United Kingdom

† Deceased.
* daniel.leightley@kcl.ac.uk

## Abstract

This study assessed the feasibility of deploying RationAI, a personalized AI-supported messaging framework, to reduce alcohol consumption among UK Armed Forces veterans. Participants were given DrinksRation, a mobile phone app, and allocated to receive either personalized or generic behavior change messages over 12 weeks. A total of 2,871 participants registered for an account during the study period. Feasibility was evaluated through recruitment (n = 2,871), retention (25.4% met engagement criteria), app usage and message delivery. Of those eligible, 343 participants were allocated to the personalized messaging group and 385 to the generic message group. The personalized group had higher early engagement, with app usage peaking at 212.4 (95% CI: 207.32 to 217.45) seconds in Week 2 compared to 183.7 seconds in the control group (95% CI: 178.90 to 188.46; p < 0.001) and received more notifications on average, reflecting additional personalized and event-triggered messages delivered as part of the intervention (47.7 [SD = 18.8] vs 16.3 [SD = 5.3]). Alcohol consumption declined in both groups over the 12-week period, with the personalized group showing a greater reduction from 31.08 to 13.20 units per week, compared to 31.24 to 15.17 units in the control group. Statistically significant between-group differences were observed at Week 2 (p = 0.027), Week 3 (p = 0.041), Week 4 (p = 0.008), and Week 10 (p = 0.049), favoring the personalized group, although between-group differences attenuated towards Week 12. Despite high attrition, the app engaged participants from an important population. These findings suggest the feasibility of personalized digital interventions for alcohol reduction, but there is a need for improved strategies to enhance long-term engagement.

which permits unrestricted use, distribution, and reproduction in any medium, provided the original author and source are credited.

**Data availability statement:** The data has also been uploaded to the Open Science Framework. This can be accessed via https://osf.io/7s6j8.

**Funding:** The author(s) received no specific funding for this work.

**Competing interests:** DL is a reservist in the UK Armed Forces. This work has been undertaken as part of his civilian employment. VC is supported by EPSRC-funded King's Health Partners Digital Health Hub (EP/X030628/1). DM is a trustee at the Forces in Mind Trust who funded an earlier trial of DrinksRation. NTF is partly funded by a grant from the UK Ministry of Defence and is a trustee (non-paid) of a charity supporting the health and wellbeing of service personnel, veterans and their families.

## Author summary

Alcohol misuse remains a major health concern among UK Armed Forces veterans, yet many experience barriers to seeking support through traditional services, including stigma, poor treatment access, and negative perceptions of care. In this study, we examined whether it was feasible to deliver RationAI, an artificial intelligence-supported messaging framework, through the DrinksRation smartphone app to veterans drinking above recommended UK guidelines. Participants received either standard generic behavior change messages or a more personalized package of messages tailored using app activity, alcohol use patterns, and self-reported wellbeing data. Over the 12-week study period, alcohol consumption declined in both groups, with the personalized group showing greater reductions during the earlier weeks of follow-up. Participants receiving personalized messages also showed higher initial app engagement and received more tailored and event-triggered notifications. However, engagement reduced over time across both groups, and many users did not remain active long enough to meet the study retention threshold. Overall, our findings suggest that AI-supported personalized messaging can be delivered feasibly to an important veteran population and may enhance early behavior change, although strategies to improve longer-term engagement are still needed.

## Introduction

There are an estimated 1.8 million Armed Forces (AF) veterans in England and Wales (United Kingdom; UK), defined by the UK Government as those serving in the military for at least 1 paid day [1], and circa 160,000 personnel currently undertaking active duty [2,3]. Among AF serving and ex-severing population, research has shown that alcohol misuse is double the rate of that seen in the UK general population [4]. To place this into context, the UK Chief Medical Officer recommends that an individual should not regularly consume more than 14 units of alcohol per week (UK standard alcohol unit equates to 8g of pure alcohol). Consuming any more than this could be harmful or hazardous to health [5]. Research has showed that alcohol misuse persists after personnel leave service [6], highlighting a need for improved care services. In the United States (US), alcohol misuse among veterans is also high, with studies showing that approximately 56.6% of veterans score for alcohol misuse, and 7.5% for problematic alcohol misuse, like patterns observed in UK veterans [7].

Research has shown that those who seek support for alcohol misuse in a clinical setting attend fewer health appointments and are more likely to have a negative perception of mental health treatment [8]. In a UK and US context, one explanation may be that those misusing alcohol are often denied access to mental health treatment services until they have reduced their hazardous drinking [9]. Therefore, interventions that target drinking behavior are needed to enhance engagement, change behaviors and promote engagement with mental health services and improve mental health outcomes and quality of life.

Since the Covid-19 pandemic, people in the UK are waiting longer for mental health referrals for alcohol use and treatment [10]. Traditional interventions such as face-to-face, drinking advice and leaflets have demonstrated some efficacy in reducing alcohol intake. However, the one-size-fits-all approach limits the ability to address individual needs in areas such as behavior, motivations, mental health and environmental factors [11]. In the UK, digital alcohol reduction has focused on the use of self-monitoring apps at a population level (e.g., Drink Less [12], Drinkaware [13]), where users are encouraged to regularly record and monitor their alcohol consumption using visualizations, dashboards and trackers [14].

Digital self-monitoring has been found to be associated with improved outcomes and is an effective Behavior Change Technique (BCT) for reducing alcohol misuse. A review of personalized digital interventions found that techniques such as behavior substitution, problem solving and providing a credible source of information, were associated with reductions in hazardous and harmful alcohol consumption [15]. Another review identified that personalized notifications, using language customized to the individual, play in a key role in promoting positive changes in behavior [16]. However, current smartphone interventions focused on the general population do not target the AF community in areas such as military language, prevailing social context, comorbid mental health problems, military service experience and perceptions of consumption [17].

One approach to addressing the lack of tailored care for AF populations is the use of artificial intelligence (AI) to personalize messaging, tailoring support to the specific individual needs, behaviors and contexts [18,19]. AI-supported interventions have the potential to enhance engagement, increase behavior change approaches, and improve health outcomes by providing timely, adaptive, and contextually relevant content which includes messages and in-app visualizations. This could be particularly beneficial to AF personnel who encounter stigma and barriers to seeking help, and digital health resources may help overcome these barriers [9,20]. By using self-monitoring, goal setting, action planning, and social support elements as part of digital interventions, AI can support engagement and habit formation and maintenance [21,22]. Behavioral data, user preferences and real-time factors can be used with AI to adjust interventions, ensuring greater relevance and effectiveness. Further, personalized interventions can deliver timely intervention prior to crisis or chronic manifestation, ensuring that behavior change prompts and supportive messages are delivered at the right moment.

This study aimed to assess the feasibility of RationAI, a personalized AI-driven messaging framework designed to reduce alcohol consumption among UK AF veterans. Feasibility was evaluated as participant engagement, app usage over time, retention and message delivery success, with preliminary analysis of behavioral outcomes. The framework was evaluated by comparing veterans who received personalized AI-supported messages against those who received generic behavior change messaging, both delivered through the DrinksRation [23] smartphone app. While DrinksRation is publicly available and can be used by any individual in the UK, it was specifically developed through co-design with the AF community to address their unique needs. For the purposes of this study, inclusion was restricted to AF veterans who met eligibility and engagement criteria. The RationAI framework integrates BCTs and adaptive user personas to tailor messages in real time, aligning content with user-reported behaviors, motivations, and mental health indicators. This adaptive approach aims to enhance adherence, reduce hazardous drinking, and support sustained behavior change.

## Methods

### Ethics

This study was approved by the King's College London Research Ethics Committee (reference: HR-19/20–17438). Informed consent was obtained from all participants. Data used in this project can be viewed on the Open Science Framework (https://osf.io/7s6j8).

### Trial design

Data used in this analysis was collected from the DrinksRation app [23–25], which is publicly available to users in the UK (efficacy of DrinksRation has been reported elsewhere [23,24]). All messages were delivered via push notifications.

Participants were allocated in the order of signup to either: 1) an intervention group receiving personalized messages, or 2) a control group without personalized messaging but receiving standard generic messaging.

## Participants

DrinksRation is available via Google Play Store or Apple iOS Store. No limits were placed on who could download or use the app. The app was promoted via social media, advertisements on social media, email and leaflets. Participants were included in this analysis if they had served in the UK AF and identified as a veteran, provided baseline demographics data, consented to data collection, consumed more than 14 units of alcohol per week at baseline and used the app for 6 weeks (based on a minimum of 3 interactions per week) out of a possible 12 weeks from baseline. 14 units of alcohol per week was selected as it is the recommended maximum consumption by the UK Chief Medical Officer. The 12-week duration was selected based on existing literature, which shows this time frame as an appropriate comparator for evaluating brief interventions, and the 6 week point for use in analysis was selected to reflect participants engaging more than half the time of the study period [26–28].

## Sample size

As this was a feasibility study, no formal sample size calculation was performed. The intent was on assessing the feasibility of recruitment over a 12-month period, during which the recruitment window remained open to observe the number of participants who could be enrolled.

## Allocation and masking

Participants were allocated using a single allocation process at the point of account creation (signup). Upon consent, each participant was assigned to a group based on a pre-determined allocation sequence (in order of sign up). No stratification or blocks were used in the allocation process. Due to the nature of the study, the research team was not blinded to participant allocation.

## Intervention

DrinksRation (formerly called *InDEx* [23,29,30]) app was developed following the Medical Research Council Complex Intervention Guidelines [31] and used a co-design approach. It was developed by the King's Centre for Military Health Research and Lancaster University, supported by experts in app development, epidemiology, addiction psychiatry, and military mental health. The app was designed as a Research Viable Product [32] to support veterans drinking at a hazardous or harmful level.

The app seeks to enhance participants' motivation and self-efficacy in modifying their alcohol consumption using BCT presented via the in-app content and through push notification messages sent to participants. The iterative development process, theoretical framework, feasibility trial, pilot and Randomized Controlled Trial are reported elsewhere [23,25,29,30].

DrinksRation was compatible with iOS and Android-supported devices released after 2017. Participants complete questionnaires via the app weekly, including questions on their mood and general mental health, which are used to personalize the app content and push notifications. Participants receive push notifications on the day that each questionnaire is due to prompt completion with further remainders every 24 hours for 3 days.

DrinksRation uses a BCT framework to promote positive changes in alcohol consumption [23,33]. The framework uses evidence-based strategies for enhancing motivation and self-efficacy. A library of generic, personalized, and personalized (event-based) message segments was developed through focus groups, stakeholder engagement, and a review of relevant literature (see Fig 1), aiming to promote adherence to DrinksRation and reduce alcohol consumption using the BCT framework [30,34,35]. Example message bank segments are available from the corresponding author.

PLOS Digital Health

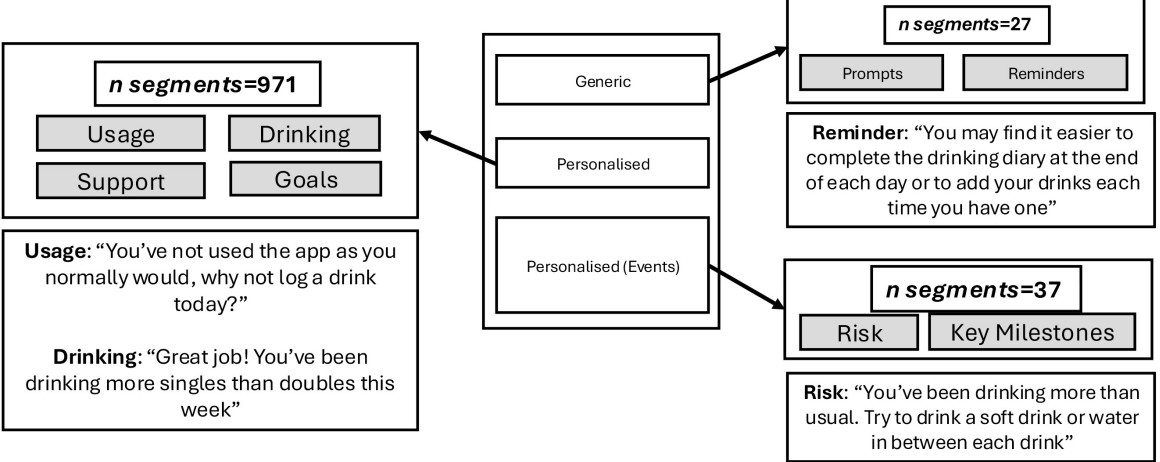

**Fig 1. Example message segments categorization.**

Both groups received the same schedule of generic notifications (including questionnaire prompts and reminders). The personalized group additionally received personalized and personalized (event) notifications generated by the RationAI framework; this higher overall notification volume is therefore a deliberate component of the intervention being evaluated rather than an unplanned imbalance between groups.

## Personalization framework

The personalization framework used Reinforcement Learning (RL) to model changes in behavior with the aim of reducing alcohol consumption via the use of push notifications. The framework is tested against adaptive user personas which are constructed for each user based on a combination of demographic information, app interaction and analytics data, alcohol consumption data, drink free day counts, and self-reported outcomes (e.g., mood, anxiety, depression and loneliness status) collected via questionnaires. This persona updates on a rolling 28-day period window. A full list of features is presented in Table 1 and is used as the input vector for the personalization framework. Active engagement is required, if a user does not engage, the adaptive persona is not generated, and generic messaging is used.

## Training

The RL model was trained using a combination of historical app usage data, synthetic data, and researcher-curated message interactions, with intended outcomes informed by user personas. The agent was a Deep Q Network (DQN) [36], mapping the adaptive persona and feature set (Table 1) to a message category decision. Synthetic data generation followed previously published approaches [37,38]. Synthetic data was generated to simulate different user engagement options. This was done to enable the model to generalize to a wide range of real-world behaviors. This approach was necessary due to the lack of training data. The training dataset included prior user engagement metrics defined in Table 1, as well as alcohol consumption trends extracted from self-reported drink diaries from prior studies. In total, the training corpus represented over 8,000 drinking patterns.

The reward function was designed around key behavioral outcomes, assigning positive reinforcement to actions that increased engagement and reduced alcohol consumption, while applying negative penalties to behaviors that led to increased alcohol consumption. Training diagnostics (average episodic reward progression and mean temporal difference loss) are provided in S1 Fig. The reward function operated as follows:

**Table 1. Feature set for modelling RationAI.**

| Feature domain | Features |
|---|---|
| Demographic information | Age, gender, service status, service length |
| App interactions and analytics data | Number of sessions, number of page views, view duration [average per day, std per day], number of times goals progress is viewed, number of messages interacted with, number of AM interactions, number of PM interactions [daily, weekly, 28-days]. This is recorded using in-app Google Analytics |
| Alcohol consumption | Weekly consumption in units, daily consumption in units, number of binge drink episodes [more than 6 units per day over 3 consecutive days], number of weekday consumption, number of weekends of consumption, number of drink free days, longest drink free streak, longest consumption streak. Alcohol consumption is measured via the drink's diary |
| Self-reported questionnaire responses | Weekly depression score, weekly anxiety score, weekly loneliness score, number of weeks with depression, number of weeks with anxiety, number of weeks with loneliness |
| Goal setting | Number of goals set, average time taken to achieve goal, number of goals missed, number of goal successes, number of engagements with app after being sent a notification related to goals [within 1 hour] |
| Notifications | Number of notifications sent, number of personalized notifications sent, number of generic notifications sent, number of personalized (event) notifications sent, average time between notification and app engagement (limited to 168 hours), number of notification engagements |
| Change trends | For all features, except 'Demographic Information', change trend features are created which present the change compared to the prior week |

- Positive reward (+1 to +10) for app interactions following a message such as opening the app within 1 hour of receiving a notification, completing a goal after receiving encouragement.

- Stronger positive reward (+10 to +20) for reductions in weekly alcohol consumption or increases drink-free days over consecutive days or weeks.

- Negative reward (-5 to -10) for disengagement after multiple notifications, preventing message fatigue, increasing consumption, or changing drinking pattern.

To prevent overfitting, the model was trained using $k$-fold cross-validation. The dataset was split into training (80%) and validation (20%) sets, where each fold used multiple iterations to evaluate performance. Hyperparameters were fine-tuned using Bayesian optimization, which iteratively adjusted parameter values to maximize the expected cumulative reward. Training progression was monitored using average episodic reward and mean temporal difference loss; early stopping was applied when validation reward plateaued.

The learning rate was optimized within a range of 0.001 to 0.1. The discount factor, which controls the weight of future rewards, was tuned between 0.7 and 0.95 to balance short-term engagement gains with long-term behavioral change. The exploration decay rate was optimized between 0.95 and 0.99, ensuring a gradual shift from exploratory behavior (randomized message selection) to exploitative behavior (prioritizing high-performing messages). Early stopping was

implemented to terminate training when improvements in reward accumulation plateaued over a predefined number of iterations. To assess learning progression and policy effectiveness, the average episodic reward was tracked throughout training, providing a performance metric for evaluating the model's ability to maximize long-term engagement and behavior change.

*Exploration versus exploitation strategy*: We adopted RL because message selection is a sequential decision problem requiring an explicit exploration/exploitation trade-off to optimize longer-term cumulative reward, rather than relying on static rule-based scheduling. The RL model seeks to balance exploration and exploitation using an e-greedy strategy with decay. Initially, a high ε value enabled broad exploration. Over time, ε decays, allowing the model to prioritize exploitation of high-performing message strategies while still testing new approaches. Thompson Sampling [39] was incorporated to refine message selection, adjusting probabilities based on the estimated effectiveness of different message types. Manual thresholding rules were developed by the research team to ensure certain types of messages did not appear together. These were deployed alongside the framework.

*Implementation*: The framework was implemented as a cloud-based service. DQN was trained offline and deployed; model parameters were not updated. Adaptation during deployment occurred through updates to the rolling 28-day adaptive persona and the exploration/exploitation strategy, with fallback to generic messaging where insufficient engagement data was present (e.g., when the user did not interact with the app). The pipeline was coded in Python, using Azure SDK for model prediction, deployment, and API engagement. The pipeline uses Azure Functions for inference with backend systems. Data storage and retrieval is managed through Google Cloud Services.

### Feasibility outcomes

In line with the aim of assessing feasibility, we examined recruitment and retention over the 12-month period, patterns of app usage (e.g., weekly interaction time) and participant responsiveness to push notifications (e.g., number received and engaged with). Retention was defined as active use of the app for at least 6 of the 12 study weeks. These feasibility indicators were chosen to inform the acceptability, usability, and practical delivery of the intervention. Participants logged their daily alcohol intake, and the app aggregated weekly consumption data. All outcome measures were collected via the DrinksRation app and stored in a cloud database. Weekly alcohol consumption and drink-free days were extracted from the drink's diary store.

### Statistical analysis

Statistical analyses were performed using STATA, and statistical significance was assessed at $p < 0.05$ where reported. Descriptive statistics were used to summarize participant characteristics at baseline and the number of notifications received over the 12-week period. Continuous variables are reported as means with standard deviations (SD), while categorical variables are presented as frequencies and percentages. No inferential statistical tests were conducted for baseline group differences [40].

The primary outcome was weekly alcohol consumption (UK units), self-reported via the in-app drinks diary. This was analyzed using a linear mixed effects model (LMM) with random intercepts for participants to account for within-subject correlation across repeated measures. Fixed effects included intervention group (personalized vs generic messaging), study week (categorical), and their interaction (group * week) to assess differential temporal effects. The model adjusted for age, gender, and history of prior alcohol treatment. Socioeconomic variables were not available in the data, and mental health symptom measures were not included in the primary outcome models due to missingness and because they may plausibly lie on the causal pathway, such that adjustment could risk over-adjustment.

Estimated marginal means with 95% confidence intervals (CIs) were calculated for each week (weeks 1–12) within each group. Between-group differences in EMMs were estimated to assess the effect of the personalized intervention over

time. Where suitable, pairwise group comparisons by week were conducted using post hoc tests with adjustment. The intent was to assess differences at each outcome point.

A secondary analysis evaluated weekly app engagement (total app usage in seconds per week), using the same LMM structure described previously comparing both groups. Fixed effects and interactions were modelled to examine group differences over time in user engagement. To support planning of a future adequately powered trial, we report week-specific between-group effect size estimates (personalized minus no personalization) with 95% CI for alcohol units and app usage in S2 Table.

The study was also reported following CONSORT (Consolidated Standards of Reporting Trials [41] and eHealth version) checklist [42].

## Results

### Demographics

Recruitment took place between May 2020 and July 2021 (see Fig 2). A total of 2871 participants were considered for inclusion. Of these, 107 (3.7%) were excluded for not providing baseline demographic data, 235 (8.2%) were excluded for not serving in the UK AF, and 409 (14.2%) were excluded for reporting alcohol consumption below 14 units per week. This resulted in 2120 (73.8%) participants being eligible for allocation, with 1218 (42.4%) assigned to the no personalization group and 902 (31.4%) to the personalization group. Following allocation, 559 (25.6%) participants in the no personalization group and 833 (39.3%) in the personalization group were excluded for having fewer than six weeks of app usage. The final analytic sample was 728 (25.4%) participants, with 385 (13.4%) in the no personalization group and 343 (11.9%) in the personalization group.

Table 2 summarizes baseline characteristics of participants. Most of the sample were male ($n$ = 631, 86.7%), served in the Army ($n$ = 556, 76.4%) and had not sought any prior treatment for alcohol use/disorder ($n$ = 636, 87.4%). Baseline characteristics by ≥6-week eligibility are provided in S2 Table.

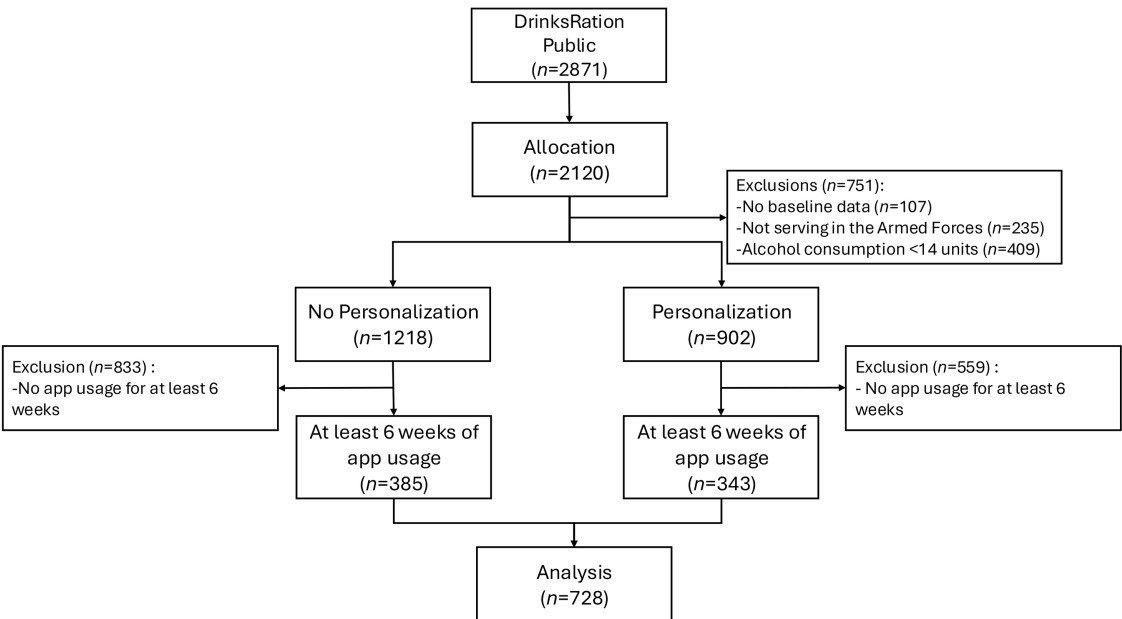

**Fig 2. CONSORT diagram for recruitment into analysis sample.**

**Table 2. Participant demographics stratified by allocation.**

| Variable | Overall (*n* = 728) | Personalization (*n* = 343) | No Personalization (*n* = 385) |
|---|---|---|---|
| Age in years (Mean, SD) | 48.4 (19.3) | 48.3 (19.2) | 48.5 (14.4) |
| **Gender (n, %)** | | | |
| Male | 631 (86.7) | 298 (86.9) | 333 (86.5) |
| Female | 97 (13.3) | 45 (13.1) | 52 (13.5) |
| **Service branch (n, %)** | | | |
| Army | 556 (76.4) | 250 (72.9) | 306 (79.5) |
| Royal Air Force | 82 (11.3) | 45 (13.1) | 37 (9.6) |
| Royal Navy | 90 (12.4) | 48 (14.0) | 42 (10.9) |
| **Phone OS (n, %)** | | | |
| iOS | 433 (59.5) | 204 (59.5) | 229 (59.5) |
| Android | 295 (40.5) | 139 (40.5 | 156 (40.5) |
| **Prior alcohol treatment (n, %)** | | | |
| No | 636 (87.4) | 293 (85.4) | 343 (89.1) |
| Yes | 92 (12.6) | 50 (14.6) | 42 (10.9) |

## Differences over time

Table 3 summarizes the estimated marginal means of weekly alcohol consumption across both groups over the 12-week periods. As shown in Fig 3, alcohol consumption declined over time in both groups. In the personalization group, consumption reduced from 31.08 units (95% CI: 29.41 to 32.74) in Week 1 to 13.20 units (95% CI: 10.23 to 16.18) by Week 12. In the no personalization group, consumption reduced from 31.24 units (95% CI: 29.66 to 32.82) to 15.17 units (95% CI: 11.76 to 18.58) over the same period.

Significant between-group differences were observed at Week 2 ($p = 0.027$), Week 3 ($p = 0.041$), Week 4 ($p = 0.008$), and Week 10 ($p = 0.049$), with the personalization group having lower alcohol unit consumption. However, the trajectories converged by Week 12, with a smaller and non-significant between-group difference at the end of follow-up ($p = 0.392$). App usage declined over time in both groups (see Table 4). In the personalized group, mean usage decreased from 207.0 seconds (95% CI: 202.0 to 212.1) at Week 1 to 116.7 seconds (95% CI: 107.2 to 126.2) at Week 12. In the no personalization group, app usage reduced from 183.9 seconds (95% CI: 179.1 to 188.8) to 104.1 seconds (95% CI: 93.2 to 115.1) over the same period. Although the personalized group initially had higher engagement, the between-group differences diminished over time. The interaction *p*-values (week 1 = 0.888, week 6 = 0.126, week 12 = 0.392) suggest that the overall rate of decline in app usage was comparable between the two groups.

## Notification differences

Table 5 summarizes the number of notifications sent to participants. This between-group difference reflects the intervention definition where generic notifications were comparable across groups, and the higher total in the personalization group is attributable to the additional personalized and event-triggered notifications. Participants in the personalization group received more notifications overall (mean = 47.7, SD = 18.8) compared to the no personalization group (mean = 16.3, SD = 5.3). While both groups received a similar number of generic notifications, those in the personalization group also received personalized (mean = 19.8, SD = 11.9) and event-triggered notifications (mean = 11.4, SD = 6.7).

**Table 3. Estimated mean weekly alcohol consumption (units) by intervention group, with 95% CI and between-group differences.**

| Week | Personalized (mean [95% CI]) | No personalization (mean [95% CI]) | Difference | p-value |
|---|---|---|---|---|
| 1 | 31.08 [29.41 to 32.74] | 31.24 [29.66 to 32.82] | -0.16 | 0.888 |
| 2 | 25.25 [23.58 to 26.92] | 27.81 [26.23 to 29.39] | -2.56 | 0.027 |
| 3 | 20.87 [19.20 to 22.54] | 25.62 [24.04 to 27.20] | -4.76 | 0.041 |
| 4 | 17.83 [16.16 to 19.50] | 20.91 [19.33 to 22.49] | -3.08 | 0.008 |
| 5 | 16.13 [14.46 to 17.80] | 18.20 [16.61 to 19.79] | -2.07 | 0.074 |
| 6 | 13.91 [12.22 to 15.59] | 15.69 [14.10 to 17.28] | -1.78 | 0.126 |
| 7 | 16.75 [14.75 to 18.75] | 17.24 [15.31 to 19.17] | -0.49 | 0.729 |
| 8 | 16.47 [14.27 to 18.66] | 18.44 [16.18 to 20.70] | -1.98 | 0.214 |
| 9 | 15.06 [12.58 to 17.53] | 16.68 [14.04 to 19.32] | -1.62 | 0.376 |
| 10 | 14.21 [11.57 to 16.85] | 18.20 [15.21 to 21.20] | -4.02 | 0.049 |
| 11 | 14.12 [11.13 to 17.11] | 17.25 [13.86 to 20.63] | -3.12 | 0.174 |
| 12 | 13.20 [10.23 to 16.18] | 15.17 [11.76 to 18.58] | -1.97 | 0.392 |

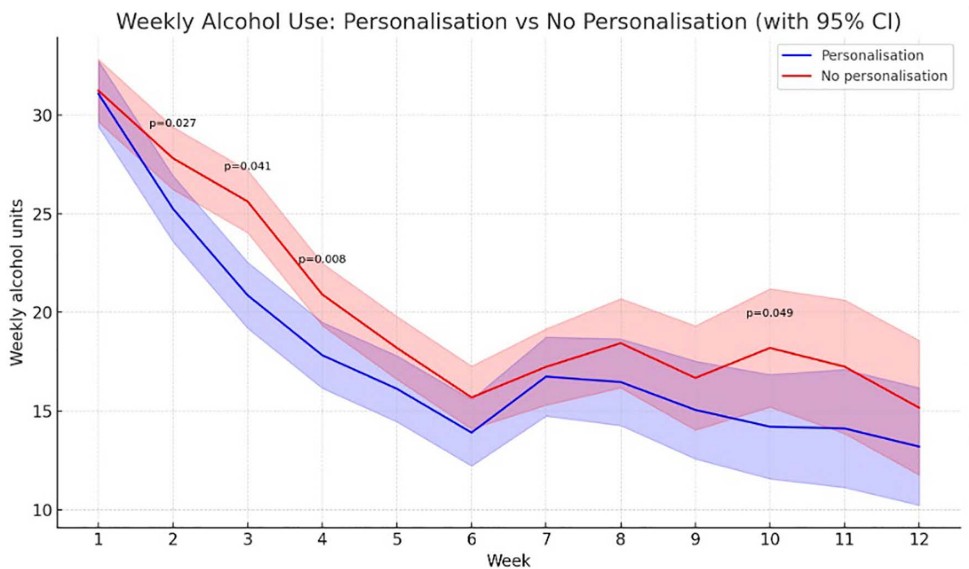

**Fig 3. Change of alcohol consumption with 95% CI provided for each week.**

## Discussion

This study evaluated the feasibility of RationAI, a personalized AI-supported messaging framework, integrated within the DrinksRation app, to support alcohol reduction among UK AF veterans. While the behavioral efficacy of the intervention has been reported in a previous RCT [23], the focus of this work is on the feasibility of delivering this personalized framework in a naturalistic real-world setting over a 12-week period compared to those who did not receive personalization.

Feasibility was defined across several dimensions, including participant engagement, retention, consistency of app usage, and delivery of personalized messaging. Of the 2,871 individuals who registered, 25.4% (n = 728) met the inclusion threshold of at least six weeks of active app use, each with a minimum of three interactions per week. This means that nearly 70% of participants did not meet this threshold, this should be considered in light a brief alcohol intervention and known long term adherence challenges [43,44].

**Table 4. Estimated mean weekly app usage (seconds) by intervention group, with 95% CI and between-group differences.**

| Week | Personalized (mean [95% CI]) | No personalization (mean [95% CI]) | Difference | p-value |
|---|---|---|---|---|
| 1 | 207.04 [201.98 to 212.10] | 177.92 [173.14 to 182.70] | 29.12 | <0.001 |
| 2 | 212.39 [207.32 to 217.45] | 183.68 [178.90 to 188.46] | 28.71 | <0.001 |
| 3 | 192.09 [187.03 to 197.15] | 178.14 [173.36 to 182.92] | 13.95 | <0.001 |
| 4 | 203.12 [198.05 to 208.19] | 179.21 [174.43 to 183.99] | 23.91 | <0.001 |
| 5 | 176.22 [171.15 to 181.29] | 169.17 [164.35 to 173.98] | 7.06 | 0.048 |
| 6 | 194.27 [189.15 to 199.40] | 171.68 [166.87 to 176.49] | 22.6 | <0.001 |
| 7 | 171.00 [164.81 to 177.20] | 165.59 [159.59 to 171.58] | 5.42 | 0.218 |
| 8 | 144.06 [137.21 to 150.91] | 155.92 [148.82 to 163.02] | -11.86 | 0.018 |
| 9 | 159.27 [151.46 to 167.07] | 150.47 [142.09 to 158.85] | 8.8 | 0.132 |
| 10 | 128.93 [120.58 to 137.29] | 130.98 [121.40 to 140.55] | -2.05 | 0.752 |
| 11 | 117.15 [107.62 to 126.68] | 101.04 [90.18 to 111.90] | 16.11 | 0.029 |
| 12 | 116.43 [106.95 to 125.90] | 103.90 [92.96 to 114.83] | 12.53 | 0.09 |

**Table 5. Descriptive on the number and type of notifications sent stratified by allocation.**

| Overall (n = 157) | Personalization (n = 343) | No Personalization (n = 385) |
|---|---|---|
| Generic (Mean, SD) | 16.5 (5.4) | 16.6 (5.3) |
| Personalized (Mean, SD) | 19.8 (11.9) | – |
| Personalized (Event) (Mean, SD) | 11.4 (6.7) | – |
| Total (Mean, SD) | 47.7 (18.8) | 16.3 (5.3) |

Among those who met the inclusion threshold, app engagement was generally high. Participants in the personalized group received and engaged with a higher volume of notifications, including tailored and event-triggered messages, compared to the control group. This indicates that the personalized framework was able to deliver messages successfully and that these messages were accessed by participants. It is important to note that app usage across the two groups remained similar. This indicates that personalization did not influence app usage.

Alcohol consumption decreased in both groups, with earlier between-group differences at several weeks but attenuation towards the end of follow-up (Week 12 difference non-significant). This pattern suggests the personalized messaging package may accelerate early reductions rather than produce a sustained separation at 12 weeks. Convergence may reflect reduced engagement over time, habituation to notifications, diminishing marginal effects of repeated prompts, and/or regression towards a stable level of consumption in both groups.

The observed drop-off in engagement, both pre-inclusion and across the 12-week period, reflects a broader challenge in digital health interventions, especially when deployed at scale and without clinician support. Future research should explore strategies to increase retention and promote sustained engagement. These may include social or peer support features, gamified elements, or embedded feedback loops to increase interactivity and perceived value [45]. Additionally, the use of adaptive notification schedules or behavioral nudges may prevent notification fatigue and maintain motivation throughout the intervention.

While this study did not aim to isolate specific behavioral mechanisms, differences between the groups suggest that personalized messaging may contribute to earlier reductions in alcohol use and support feasibility of delivery in a real-world setting, although sustained separation at 12 weeks was not observed. These findings build upon prior evidence that personalization and timing are critical components of digital behavior change interventions [15,16,21]. In this study,

personalization was implemented via a machine learning algorithm informed by adaptive user personas. The successful deployment and real-time operation of this system support its feasibility and scalability for broader use.

There are several limitations of this work. First, because the AI-supported intervention comprises both tailored content and an expanded notification strategy (including event-triggered delivery), this feasibility study cannot disentangle the independent contribution of personalization content versus message dose; a factorial design would be required to isolate these components. Second, the inclusion criteria meant that only a subset of all app users were analyzed, and these individuals may differ systematically from those who disengaged early. While this is common in digital intervention studies [46], it raises questions about generalizability. Third, alcohol consumption was self-reported via in-app drink diaries, which may be subject to recall or social desirability bias. Future studies may consider integrating passive monitoring tools, such as wearable sensors or linked breathalyzer data, to support objective outcome measurement. Fourth, although this feasibility study was not powered to detect subgroup effects, further work is needed to understand how different groups, by gender, age, or comorbidity, engage with and respond to personalized interventions. The sample was drawn from a real-world population of veterans, many of whom likely experience complex comorbidities, including mental health problems, which are known to affect treatment engagement [8,9,17]. Finally, the absence of financial incentives and the fully remote delivery model may also have contributed to higher attrition. Nevertheless, the fact that over 700 participants remained engaged at a meaningful level, interacting with the app regularly over multiple weeks, suggests that the intervention was acceptable to a substantial subset of users.

Despite these limitations, the study demonstrates that approximately 30% of veterans were willing to engage with a digital intervention aimed at reducing alcohol consumption, and that nearly half of those remained engaged throughout the intervention period. Given the barriers to help-seeking in the veteran community [9,47–50], this level of engagement is encouraging. Our findings suggest that AI-supported messaging strategies hold promise as scalable, acceptable tools for delivering behavior change support in populations with high levels of unmet need. Future research should focus on refining personalization algorithms, extending duration of follow-up, and testing strategies to support longer-term engagement and behavior change.

These findings have implications for the design and implementation of digital health interventions targeting alcohol use in the AF community. Generic messaging, while useful for broad implementation, does not adequately address the complex social, psychological and occupational factors influencing alcohol consumption in veteran. AI supported models, provide a data-driven method for optimizing message delivery based on real-world user behavior. The integration of AI supported personalization frameworks allows for more precise, adaptive, and user focused interventions that can better address individual needs and behavioral patterns. Given the high prevalence of alcohol misuse in the AF, such approaches could provide a scalable and cost-effective alternative to conventional interventions. This could include in areas such as common mental health disorders [51], gambling [52,53] and PTSD [54,55] interventions which do not currently have any personalization approaches tailored to high intensity occupations such as the military.

This study provides preliminary evidence supporting the use of AI supported personalized messaging for reducing alcohol consumption in the UK AF. By integrating behavioural data, adaptive user personas, and Just-In-Time interventions, AI can enhance engagement and optimize behaviour change strategies beyond conventional static interventions. These findings highlight the potential of AI-supported digital health solutions in tackling alcohol misuse, particularly in populations facing barriers to traditional care. Future research should focus on long-term efficacy, objective outcome validation, and scalability.

## Supporting information

**S1 Fig. Training results.**
(DOCX)

**S1 Table. Baseline characteristics of ≥6 weeks vs < 6 weeks.**
(DOCX)

**S2 Table. Effect size estimates.**
(DOCX)

## Acknowledgments

We note with great sadness the passing of our co-author, Professor Nicola T. Fear, before the final version of this publication. Professor Fear made an important contribution to this work, and we are grateful for her scholarship, guidance, and enduring impact on the field. Dr Daniel Leightley accepts responsibility for the integrity and validity of the data collected and analyzed.

## Author contributions

**Conceptualization:** Daniel Leightley, Laura Goodwin.

**Data curation:** Charlotte Williamson.

**Formal analysis:** Daniel Leightley.

**Investigation:** Daniel Leightley, Roberto J Rona, Dominic Murphy, Nicola T. Fear, Laura Goodwin.

**Methodology:** Daniel Leightley, Roberto J Rona, Dominic Murphy, Nicola T. Fear, Laura Goodwin.

**Project administration:** Charlotte Williamson.

**Software:** Daniel Leightley.

**Writing – original draft:** Daniel Leightley.

**Writing – review & editing:** Daniel Leightley, Charlotte Williamson, Iain J. Marshall, Vasa Curcin, Roberto J Rona, Dominic Murphy, Nicola T. Fear, Laura Goodwin.

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
