## [Decision Letter · Decision Letter 0]

15 Dec 2025

Response to Reviewers'. This file does not need to include responses to any formatting updates and technical items listed in the 'Journal Requirements' section below.'. This file does not need to include responses to any formatting updates and technical items listed in the 'Journal Requirements' section below.* A marked-up copy of your manuscript that highlights changes made to the original version. You should upload this as a separate file labeled 'Revised Manuscript with Track Changes'.'.* An unmarked version of your revised paper without tracked changes. You should upload this as a separate file labeled 'Manuscript'.'. If you would like to make changes to your financial disclosure, competing interests statement, or data availability statement, please make these updates within the submission form at the time of resubmission. Guidelines for resubmitting your figure files are available below the reviewer comments at the end of this letter. We look forward to receiving your revised manuscript. Kind regards, Benjamin FiltjensAcademic EditorPLOS Digital Health Leo Anthony CeliEditor-in-ChiefPLOS Digital Healthorcid.org/0000-0001-6712-6626 **Journal Requirements:** If the reviewer comments include a recommendation to cite specific previously published works, please review and evaluate these publications to determine whether they are relevant and should be cited. There is no requirement to cite these works unless the editor has indicated otherwise.  **Additional Editor Comments (if provided):****Reviewers' Comments:** Reviewer's Responses to Questions

**Comments to the Author**

1. Does this manuscript meet PLOS Digital Health’s publication criteria? Is the manuscript technically sound, and do the data support the conclusions? The manuscript must describe methodologically and ethically rigorous research with conclusions that are appropriately drawn based on the data presented.? Is the manuscript technically sound, and do the data support the conclusions? The manuscript must describe methodologically and ethically rigorous research with conclusions that are appropriately drawn based on the data presented.

Reviewer #1: Partly

Reviewer #2: Yes

2. Has the statistical analysis been performed appropriately and rigorously?

Reviewer #1: I don't know

Reviewer #2: Yes

3. Have the authors made all data underlying the findings in their manuscript fully available (please refer to the Data Availability Statement at the start of the manuscript PDF file)?

The PLOS Data policy requires authors to make all data underlying the findings described in their manuscript fully available without restriction, with rare exception. The data should be provided as part of the manuscript or its supporting information, or deposited to a public repository. For example, in addition to summary statistics, the data points behind means, medians and variance measures should be available. If there are restrictions on publicly sharing data—e.g. participant privacy or use of data from a third party—those must be specified.requires authors to make all data underlying the findings described in their manuscript fully available without restriction, with rare exception. The data should be provided as part of the manuscript or its supporting information, or deposited to a public repository. For example, in addition to summary statistics, the data points behind means, medians and variance measures should be available. If there are restrictions on publicly sharing data—e.g. participant privacy or use of data from a third party—those must be specified.

Reviewer #1: No

Reviewer #2: Yes

4. Is the manuscript presented in an intelligible fashion and written in standard English?

Reviewer #1: Yes

Reviewer #2: Yes

Reviewer #1: The paper presents a feasibility study for RationAI, a personalized AI-based message recommendation system, integrated with the DrinksRation mobile phone app.

The goal of the AI framework is to increase the effectiveness of the DrinksRation app, with a study conducted among UK Armed Forces veterans.

Given my profile, I will mainly comment on the AI part. In general the paper is well written, however I have two main concerns.

1. Table 5 highlights that the `No Personalization` group received much fewer notifications w.r.t. the `Personalization` group. This creates a problem: is the higher effectiveness in lowering alcohol consumption seen in the `Personalization` group given by the PERSONALIZATION of messages, or simply by the higher amount of received messages?

Unfortunately, this makes it difficult to assess the effectiveness of the RL model.

2. The training and deployment of the RL model can be detailed more. What's the overall size of the training set? Are the more details on the used synthetic data, as well as the researcher-curated data? Is there any training metric available, like loss, etc? Which specific RL algorithm is used and: is RL really necessary in this setting? Was the model adapted during deployment, or was it trained completely offline before the actual use?

Reviewer #2: The authors intend to address an important public health issue on alcohol misuse among UK Armed Forces veterans by evaluating the feasibility of an AI-driven personalized messaging framework integrated into a mobile app. The study is innovative by leveraging reinforcement learning for personalization. The paper is overall well-written and structured. While the authors provide strong technical rigor and clear reporting, there are some limitations in the method, and the results need to be addressed.

1) There is a lack of description on the AI personalization algorithm performance metrics, for example, accuracy, reward progression.

2) It is unclear why there is no comparison between the characteristics of participants who dropped out and those who completed at least 6 weeks of app usage.

3) It is unclear why known factors such as mental health comorbidities and socioeconomic factors impacting alcohol usage were not included and adjusted beyond age, gender, and prior treatment.

4) It would be interesting to include subgroup analyses (for example, gender, age, prior treatment) to explore differential effects.

5) While the authors reported significant differences at some weeks, the overall trend converges by Week 12. This needs to be further discussed.

6) It would be beneficial to report effect sizes for the results, which is necessary for future large clinical trials to estimate the sample size.

7) Are there any data on user satisfaction or perceived relevance of messages? It would be helpful to examine the qualitative feedback on the app usage or non-use.

**Do you want your identity to be public for this peer review?** For information about this choice, including consent withdrawal, please see our Privacy Policy..

Reviewer #1: No

Reviewer #2: **Yes**

**Figure resubmission:**  While revising your submission, we strongly recommend that you use PLOS’s NAAS tool (https://ngplosjournals.pagemajik.ai/artanalysis) to test your figure files. NAAS can convert your figure files to the TIFF file type and meet basic requirements (such as print size, resolution), or provide you with a report on issues that do not meet our requirements and that NAAS cannot fix. 

**Reproducibility:** To enhance the reproducibility of your results, we recommend that authors of applicable studies deposit laboratory protocols in protocols.io, where a protocol can be assigned its own identifier (DOI) such that it can be cited independently in the future. Additionally, PLOS ONE offers an option to publish peer-reviewed clinical study protocols. Read more information on sharing protocols at https://plos.org/protocols?utm_medium=editorial-email&utm_source=authorletters&utm_campaign=protocols To enhance the reproducibility of your results, we recommend that authors of applicable studies deposit laboratory protocols in protocols.io, where a protocol can be assigned its own identifier (DOI) such that it can be cited independently in the future. Additionally, PLOS ONE offers an option to publish peer-reviewed clinical study protocols. Read more information on sharing protocols at https://plos.org/protocols?utm_medium=editorial-email&utm_source=authorletters&utm_campaign=protocols

---

## [Decision Letter · Decision Letter 1]

8 Mar 2026

Reducing alcohol consumption in UK Armed Forces veterans: feasibility of using personalized push notifications using AI

PDIG-D-25-00448R1

Dear Dr. Leightley,

We are pleased to inform you that your manuscript 'Reducing alcohol consumption in UK Armed Forces veterans: feasibility of using personalized push notifications using AI' has been provisionally accepted for publication in PLOS Digital Health.

Best regards,

Benjamin Filtjens

Academic Editor

PLOS Digital Health

**Additional Editor Comments (if provided):**

**Reviewer Comments (if any, and for reference):**

Reviewer's Responses to Questions

**Comments to the Author**

Reviewer #1: All comments have been addressed

publication criteria? Is the manuscript technically sound, and do the data support the conclusions? The manuscript must describe methodologically and ethically rigorous research with conclusions that are appropriately drawn based on the data presented.? Is the manuscript technically sound, and do the data support the conclusions? The manuscript must describe methodologically and ethically rigorous research with conclusions that are appropriately drawn based on the data presented.

Reviewer #1: Yes

3. Has the statistical analysis been performed appropriately and rigorously?

Reviewer #1: Yes

4. Have the authors made all data underlying the findings in their manuscript fully available (please refer to the Data Availability Statement at the start of the manuscript PDF file)?

The PLOS Data policy requires authors to make all data underlying the findings described in their manuscript fully available without restriction, with rare exception. The data should be provided as part of the manuscript or its supporting information, or deposited to a public repository. For example, in addition to summary statistics, the data points behind means, medians and variance measures should be available. If there are restrictions on publicly sharing data—e.g. participant privacy or use of data from a third party—those must be specified.requires authors to make all data underlying the findings described in their manuscript fully available without restriction, with rare exception. The data should be provided as part of the manuscript or its supporting information, or deposited to a public repository. For example, in addition to summary statistics, the data points behind means, medians and variance measures should be available. If there are restrictions on publicly sharing data—e.g. participant privacy or use of data from a third party—those must be specified.

Reviewer #1: Yes

5. Is the manuscript presented in an intelligible fashion and written in standard English?

Reviewer #1: Yes

Reviewer #1: (No Response)

**Do you want your identity to be public for this peer review?** For information about this choice, including consent withdrawal, please see our Privacy Policy..

Reviewer #1: No
